# MM-Forecast: A Multimodal Approach to Temporal Event Forecasting with Large Language Models

## ABSTRACT

We study an emerging and intriguing problem of multimodal temporal event forecasting with large language models. Compared to using text or graph modalities, the investigation of utilizing images for temporal event forecasting has received less attention, particularly in the era of large language models (LLMs). To bridge this gap, we are particularly interested in two key questions of: 1) why images will help in temporal event forecasting, and 2) how to integrate images into the LLM-based forecasting framework. To answer these research questions, we propose to identify two essential functions that images play in the scenario of temporal event forecasting, *i.e.*, **highlighting and complementary**. Then, we develop a novel framework, named **MM-Forecast**. It employs an Image Function Identification module to recognize these functions as verbal descriptions using multimodal large language models (MLLMs), and subsequently incorporates these function descriptions into LLM-based forecasting models. To evaluate our approach, we construct a new multimodal dataset, MidEast-TE-mm, by extending an existing event dataset MidEast-TE with images. Empirical studies demonstrate that our MM-Forecast can correctly identify the image functions, and further more, incorporating these verbal function descriptions significantly improves the forecasting performance. The dataset, code, and prompt will be released upon acceptance.

## CCS CONCEPTS

• **Information systems** → **Multimedia and multimodal retrieval**; • **Computing methodologies** → **Temporal reasoning**.

## KEYWORDS

Temporal Event Forecasting, Multimodal Event Forecasting

## 1 INTRODUCTION

Temporal event forecasting aims to predict future events according the observed events in history. The forecasting of critical events, such as pandemic outbreak, civil unrest, and international conflicts, can help shape policies in advance and minimize potential impacts. Due to its great potential application value, temporal event forecasting [5, 14, 20, 26, 27, 29] has garnered increasing attention from both the academic and industrial community in recent years. Despite

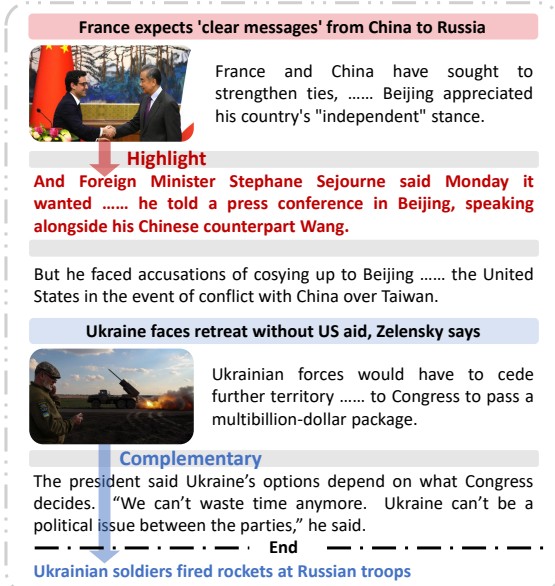

**Figure 1: Illustration of our motivation about why images will help in temporal event forecasting. We identify two essential functions of images, *i.e.,* highlighting and complementary. By offering auxiliary highlighting or complementary information, images enhance the understanding of temporal events, thus boosting the forecasting performance.**

promising progress, current methods have ignored the rich multimodal information, *e.g.,* images, remaining to be an unexplored research gap.

With the enormous success of Large Language Models (LLMs), an increasing number of studies [16, 22, 25, 38] have been exploring LLMs for the task of temporal event forecasting to enhance the forecasting accuracy. These pioneering works explore the application of LLMs in the task of temporal event forecasting, leveraging techniques such as in-context learning (ICL) [16], instruction tuning [25, 38], and retrieval-augmented generation (RAG) [33]. Compared to traditional methods, LLM-based methods offer several advantages in terms of effectiveness, flexibility, and scalability. Traditional non-LLM methods [15, 20, 27, 29], whether based on structured or unstructured data, typically require large-scale well-annotated datasets. Moreover, model selection is often a challenge for these traditional techniques due to high computational costs. Additionally, traditional methods generally require separate training for different datasets, as a result, they often struggle to make fast adaptation *w.r.t.* frequent changing in dataset and temporal shifts. Therefore, the application of LLMs to the task of temporal event forecasting holds significant potential and promise [16]. However, all of the existing LLM-based methods only consider a single modality, such as text [16] or graph [25], while ignoring the

prevalent visual modality, *i.e.,* images. Some previous works have justified that images are helpful in multimodal event detection and extraction [19, 36], while none of them investigate images' utility in temporal event forecasting.

To bridge this gap, we aim to integrate images into temporal event forecasting and construct multimodal temporal event forecasting models. However, it is a non-trivial objective due to the following challenges. First, it is necessary to investigate the function between visual information and other modal information, *i.e.,* the interplay between visual and textual modalities. Next, we need to figure out how this function between the two modalities can contribute to the task of temporal event forecasting. Second, while prior work [36] has explored the image function in related tasks such as event extraction, these approaches typically require large amounts of labeled training data. Additionally, they often struggle to generalize effectively to other task definitions. Therefore, there is a pressing need to design an effective method to identify the function between modalities and seamlessly integrating them into LLM-based forecasting models.

To address the aforementioned issues, we propose a novel framework for multimodal temporal event forecasting, named as **MM-Forecast**. Specifically, we identify two essential functions of images, *i.e.,* the highlighting and complementary. As illustrated in Figure 1, when the function of associated image is highlighting, the image serves to emphasize key events. In contrast, when the function of associated image is complementary, the image provides supplementary information that complements the textual content. In order to recognize these two types of functions, we propose an Image Function Identification module that is based on Multimodal LLMs (MLLMs) due to their superior multimodal understanding and reasoning capabilities in zero-shot settings. The proposed module is designed to recognize the function of images in historical events, and then transform this information into verbal descriptions that can be seamlessly integrated into the LLM-based event forecasting model. To demonstrate the scalability of our approach, we have integrated it into two distinct LLM-based forecasting models, *i.e.,* one based on the in-context learning (ICL) method [16], and the other based on the retrieval-augmented generation (RAG) technique [17]. In order to evaluate our approach, we construct an exploratory dataset by engaging images into an existing dataset MidEast-TE [27]. We name this new dataset MidEast-TE-multimodal (short as **MidEast-TE-mm**). In the final evaluation, with the enhancement of visual information, the temporal event forecasting task achieves superior forecasting accuracy compared to the unimodal approach. The experimental results illustrate that our method accurately recognizes the function of images in various aspects. Furthermore, the findings demonstrate that multimodal temporal forecasting represents a potential and promising research direction worthy of further exploration. The main contributions are as follows:

- To the best of our knowledge, this is the first comprehensive investigation into the integration of visual information for temporal event forecasting in the era of LLMs.
- We identify the function of images within the context of temporal event forecasting, and design an overall framework to recognise and integrate these visual information into LLM-based forecasting models.

- Extensive experiments illustrate that our framework accurately identifies the function of images and demonstrate that visual information can enhance the performance of temporal event forecasting. Furthermore, these findings have led to several noteworthy and valuable directions for future research.

## 2 RELATED WORKS

The related works in this paper are surveyed from two perspectives: existing approaches to temporal event forecasting, and the application of large language models (LLMs) and multimodal LLMs (MLLMs) for event analysis.

### 2.1 Temporal Event Forecasting

Temporal event forecasting centers on predicting future event occurrences based on the historical events. The existing approaches can be classified into three main paradigms based on the event format: time series, structured events, unstructured events.

For the time series paradigm, existing works [2, 21, 28] typically represent events as an ordered sequence of data points that describe the progression of actions or occurrences. However, this approach inherently fails to represent multiple relationships between entities. Alternatively, another branch of works [8, 32, 34, 39] focus on the prediction of structured events, *i.e.,* using graph to represent events, which is known as temporal knowledge graph (TKG). Representative TKG methods [15, 20, 29] extend the static knowledge graph completion techniques, aiming to learn and aggregate the temporal and relational patterns among entities for forecasting. Recent works[27] also introduce the context into the temporal event forecasting, elaborating the event's occurrence situation or condition. Some other works, such as RESIN-11 [10] and IED [18], represent temporal event with pre-defined complex event schema. In addition, several studies have explored the use of unstructured textual representations of temporal events, where each atomic event is generated from multi-document summaries [11] or event chains [13]. However, all of them still conduct the forecast reasoning on single modality data only. Some works [19, 36] explore the image function in event extraction task, while none of them investigate images' utility in temporal event forecasting.

### 2.2 LLMs for Event Analysis

The tremendous success of large language models (LLMs) in recent years, exemplified by GPT-3 [4] and its numerous successors [6, 7, 37, 40], has inspired researchers to explore the application of these powerful models to various event-related tasks. While a significant portion of existing work has focused on temporal event understanding rather than forecasting, a few studies have leveraged LLMs for the task of temporal event forecasting. Specifically, the GPT-NeoX-ICL [16] method and GENTKG [22] method have explored the use of LLMs for event forecasting. The former leverages in-context learning of LLMs and constructs prompts as a list of historical events each in quadruplet format, while the latter improves the selection of historical event inputs by a temporal logical rule-based retrieval strategy. However, these existing LLM-based methods still rely solely on single-modality data, potentially missing valuable information from other modalities, such as images. With the success of LLMs, MLLMs, such as Flamingo [1], LLaVA [23],

and Gemini [35], have emerged as promising means for integrating visual and textual modalities. These MLLMs have demonstrated impressive performance across various visual-language tasks, suggesting their potential for enhancing temporal event forecasting by leveraging visual information. Therefore, our work focuses on multimodal temporal event forecasting with LLMs.

## 3 OUR APPROACH: MM-FORECAST

The overall framework of our proposed approach is depicted in Figure 2. We first formally define the multimodal temporal event forecasting task in Section 3.1. Second, we specifically introduce the key module of Image Function Identification in Section 3.2. Finally, we elaborate on how to integrate the recognized image functions into LLM-based forecasting models in Section 3.3.

### 3.1 Problem Formulation

To give formal definition of the problems, we separate it into two sub-tasks given the different data representation of historical information. Detailed definitions and qualitative examples, such as complex events (CE), are presented by the supplementary material. **Structured Event Forecasting (Graph[1]).** Structured data-based methods typically define each event as a quadruple $(s, r, o, t)$, which is also called an atomic event, where $s, r, o, t$ corresponds to the subject, relation, object, and timestamp. At each timestamp $t$, all the quadruples form an event graph, denoted as $G_t = \{(s, r, o, t)\}^N$, where $N$ is the number of events in timestamp $t$. Recent works[27] have further introduce the complex event (CE) into the structured event representation by document clustering techniques, elaborating the event's occurrence situation or context. Specifically, each historical event is extended from a quadruple to a quintuple, *i.e.,* $(s, r, o, t, c)$, where $s \in \mathcal{E}, r \in \mathcal{R}, o \in \mathcal{E}$, and $c \in C$ represent the subject, relation, object, and CE, respectively; $\mathcal{E}, \mathcal{R}$ and $C$ are the entity set, relation set and context set. Correspondingly, the event graph at each timestamp will be extended as $G_t = \{(s, r, o, t, c)\}^N$. The overall structured event forecasting task can then be formulated as follows: Given the historical event graphs $G_{<t} = \{G_0, G_1, ..., G_{t-1}\}$ before timestamp $t$, and a query $(s, r, t)$ or $(s, o, t)$, the goal is to predict the missing object or relation.

**Unstructured Event Forecasting (Text[2]).** In addition to the structured event representation, we also consider the unstructured representation of historical events, where the historical information is provided in the form of textual sub-events, *i.e.,* $A_t = [a_1, a_2, ..., a_k]_{k=1}^K$ and $A_t \in \mathcal{A}$, where $a_k$ denotes the k-th textual sub-events and $\mathcal{A}$ denotes the corpus of textual sub-events. The textual sub-events are obtained by summarizing the content of news articles. The unstructured event forecasting task can be formulated as: Given the historical textual sub-events $A_{<t} = \{A_0, A_1, ..., A_{t-1}\}$ before timestamp $t$, and a query $(s, r, t)$ or $(s, o, t)$, the goal is to predict the missing object or relation.

### 3.2 Image Function Identification

In news articles, images play a vital role not only in attracting readers but also in completing and enriching the textual content. We

will identify the image functions into three categories, *i.e.,* highlighting, complementary, and irrelevant, by MLLMs during the dataset construction stage.

Excluding the irrelevant images, the others serve distinct roles in the temporal event forecasting task. We propose an Image Function Identification module to recognize these functions as verbal descriptions using MLLMs, and subsequently incorporates these function descriptions into LLM-based forecasting models. Specifically, when the function of associated image is highlighting, the visual elements directly support and highlight the key sub-events described in the text. These "highlighting" sub-events, substantiated by corroborating information across modalities, can be identified as key events. To determine which sub-event is a key event, we leverage the multimodal large language models (MLLMs) to analyze the images and sub-events along multiple dimensions, including main objects, celebrities, activities, environment, and labeled items. In cases where the function of associated image is complementary, the visual content contains information that supplements and extends beyond what is covered in the news text. To more effectively extract the relevant supplementary information, we consider the following aspects: 1) Identify the main subject of the image as the central point. 2) Directly relate the extracted information to the news event in the article. 3) Prioritize the most newsworthy visual elements. 4) Ensure all information comes directly from the provided news article without fabrication, and 5) Aim for a concise summary using clear language. By analyzing the interplay between visual images and textual content within news articles, we can gain a more comprehensive understanding of the underlying events and better contextualize the temporal progression of historical events. This multimodal approach, which leverages both linguistic and visual modalities, holds the promise of enhancing the accuracy of temporal event forecasting. Ultimately, the prompts utilized in making predictions are shown below:

```
SYSTEM:
You are an assistant to perform event forecasting
with the following rules:
1. The atomic event is the basic unit describing a spec-
ific event, typically presented in the form of a quadru-
ple (S, R, O, T), where S represents the subject, R repre-
sent the relation, O represents the object, and T repres-
ents the relative time.
2. When formulating the ultimate prediction, the preemi-
nent factor to be meticulously weighed and scrutinized
is the [Key Events]. Complementing this paramount consi-
deration is the [Related events], which, though ancilla-
ry in nature, serves as a valuable adjunct, furnishing
pertinent contextual details and auxiliary insights to
fortify the predictive analysis.
3. Given a query of (S, R, T) in the future and the list
of historical events until t, event forecasting aims to
predict the missing object.
USER:
[Query]: (S, O/R, T)
[Key Events]: xxx.
[Related Events]: xxx.
[Options]: A.xxx B.xxx C.xxx D.xxx E.xxx
```

---

[1]"Graph" is interchangeably used to represent this setting.

[2]"Text" is interchangeably used to represent this setting.

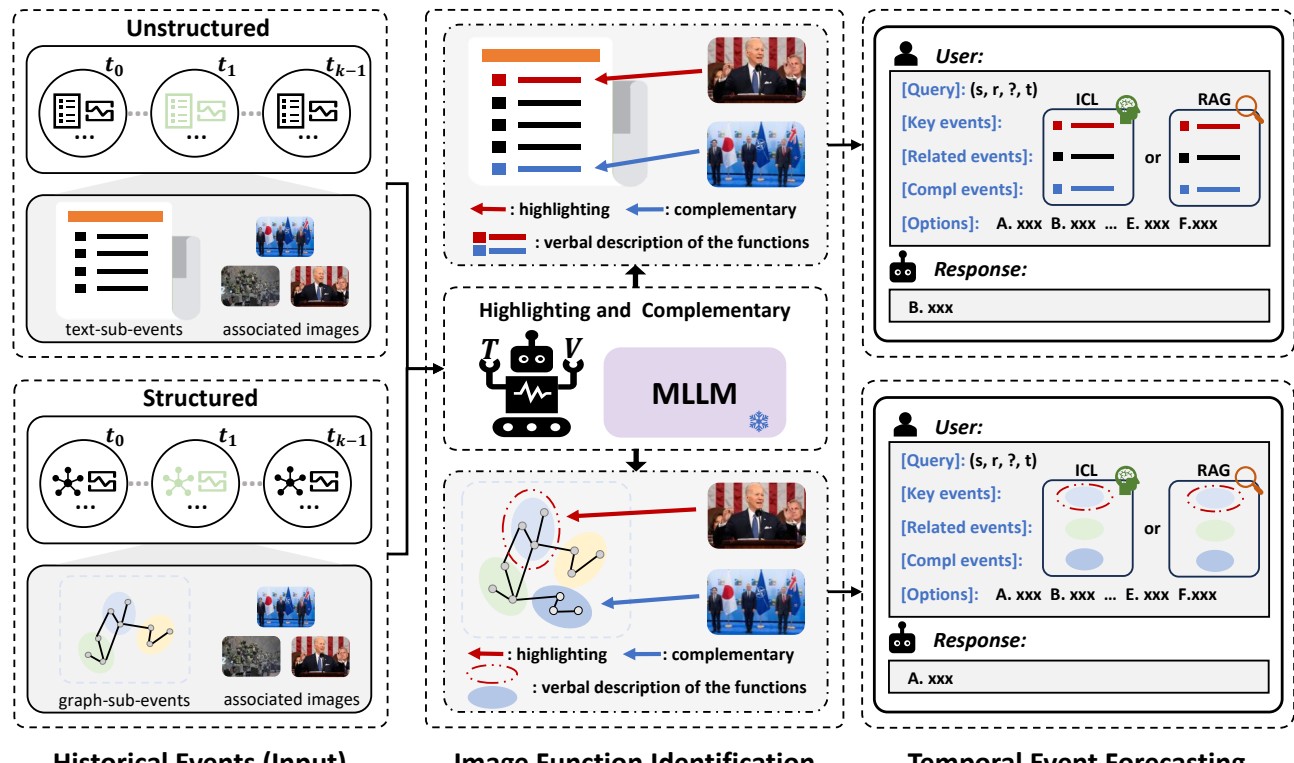

**Figure 2: The schematic overview of MM-Forecast. By consuming historical events in either format of unstructured or structured input (left), our image function identification module (middle) recognizes the image functions as verbal descriptions, which are then feed into LLM-based forecasting model (right). Our framework is versatile to handle both structured and unstructured events, meanwhile, it is compatible to popular LLM components for event forecasting, *i.e.,* ICL and RAG.**

The key events are explicitly highlighted within the prompt, while complementary information is provided as additional relevant events.

## 3.3 Forecasting Framework

We follow the emerging solution [16] and leverage LLMs as the forecasting backbone. Given there are few established studies of using LLMs for event forecasting, we implement two forecasting methods by considering two representative approaches, *i.e.,* In-context Learning (ICL) [16] and Retrieval Augmented Generation (RAG) [17]. Each of these two methods can accept both structured and unstructured historical input, and answer the structured forecasting questions.

*3.3.1 In-context Learning (ICL).* In-context learning leverages both intrinsic and extrinsic factors to construct historical events. Specifically, the intrinsic factors of an event are related to its inherent elements, particularly the subject. In contrast, the extrinsic factors are driven by the contextual environment surrounding the event. Therefore, whether the data is structured or unstructured, we construct the historical events based on the subject and the complex event, separately. The details are as follows:

- **Structured Data.** For structured data, the method takes the discrete event graph as the input. To capture the intrinsic factors,

we use the subject of the current event as a guiding clue to construct the historical event graph $\mathbf{G}^s_{<t} = \{G^s_0, G^s_1, ..., G^s_{t-1}\}$, where $G^s_t$ represents historical events graph at timestamp $t$ with the same subject as the current event. To account for the extrinsic factors, we construct the historical event graph from the complex event, *i.e.* $\mathbf{G}^c_{<t} = \{G^c_0, G^c_1, ..., G^c_{t-1}\}$, where $G^c_t$ represents historical events graph at timestamp $t$ with the same complex event as the current event. Finally, with the highlighting and complementary functions of the images, the input historical event graph is $\mathbf{G}_{input} = [\mathbf{G}_k, \mathbf{G}_r, \mathbf{G}_c]$, where $\mathbf{G}_{input} \in \mathbf{G}^s_{<t} \bigcup \mathbf{G}^c_{<t}$ and $\mathbf{G}_k$ denotes the key events, $\mathbf{G}_r$ represents the remaining events, and $\mathbf{G}_c$ corresponds to the complementary events, respectively.

- **Unstructured Data.** For unstructured data, the method takes the textual sub-events as input. Firstly, we identify the events by the historical events graph from the subject and complex event and find the corresponding textual sub-events set $\mathbf{A}^s_{<t} = \{A^s_0, A^s_1, ..., A^s_{t-1}\}$ and $\mathbf{A}^c_{<t} = \{A^c_0, A^c_1, ..., A^c_{t-1}\}$ through the relationships between textual sub-events and graph sub-events. Then, with the highlighting and complementary functions of the images, the input historical textual sub-events are similarly $\mathbf{A}_{input} = [\mathbf{A}_k, \mathbf{A}_r, \mathbf{A}_c]$, where $\mathbf{A}_{input} \in \mathbf{A}^s_{<t} \bigcup \mathbf{A}^c_{<t}$ and $\mathbf{A}_k$ denotes the key events, $\mathbf{A}_r$ represents the remaining events, and $\mathbf{A}_c$ corresponds to the complementary events, respectively.

*3.3.2 Retrieval Augmented Generation (RAG).* Despite the rich information provided by in-context learning methods, the inherent nature of the temporal event means that the existing historical event still contains substantial noise. Inspired by the recent research of RAG [17], we also adopt the retrieve-then-generate paradigm to find the most relevant historical events to mitigate the problem of noise. Similar to ICL methods, we utilize two forms of data representation, structured data and unstructured data:

- **Structured Data.** Due to the structured nature of the data representation, the event graphs adhered to a unified quintuple format. Therefore, we first retrieve the entities that have interacted with the subject of the query event. Once we have obtained the related entity set, we can construct the history with the historical events where the subject or object is within this set. Similarly, through the function of images, the retrieval process also contains key events and complementary events.
- **Unstructured Data.** Unlike structured data, we can use the embedding techniques to directly retrieve relevant news events from a set of historical news articles for the unstructured data. Following this, we filter historical news events based on timestamps, eliminating outdated and irrelevant events. We also select the key events and complement information based on the images, which will be input according to the prompt described in Section 3.2, and finally obtain the prediction results.

## 4 EXPERIMENTS

We conduct experiments on our constructed MidEast-TE-mm dataset to evaluate the proposed approach and answer the following research questions:

- **RQ1:** What is the overall performance of temporal event forecasting methods with visual information?
- **RQ2:** How do the highlighting and complementary function of images affect the performance?
- **RQ3:** Is the highlighting and complementary function of images really useful?
- **RQ4:** How do different LLM backbones as well as fine-tuning affect the performance?

### 4.1 Dataset

We briefly introduce the data source and construction of the dataset, and more details of the construction, dataset statistics, and thorough evaluation of the dataset are presented in the supplementary file.

*4.1.1 Data Source.* We follow a previous dataset MidEast-TE [27] to build our dataset, named as MidEast-TE-multimodal (MidEast-TE-mm). The original MidEast-TE dataset extracts atomic events from news articles utilizing the Vicuna model [6], and identifies different complex events through clustering methodology. Given the large scale of MidEast-TE, we sample a subset of complex events from MidEast-TE and build our dataset.

*4.1.2 Dataset Construction.* The dataset construction pipeline consists of two consecutive components: sub-event extraction and image collection.
**Sub-event Extraction.** We conduct event extraction for both structured and unstructured events using LLMs. For structured data, we adopt a hierarchical extraction pipeline based on the original

dataset [27] and the three-layer structure of the CAMEO ontology [3]. Each layer of event extraction is based on the results of the prior layer to reduce cost and performance degradation due to extensive number of event types. For unstructured data, we summarize the news articles to generate multiple sub-events, ensuring accurate, comprehensive, and coherent content selection and description.
**Image Collection.** The web page of each news url in MidEast-TE is associated with one or more images, which can be used as the visual information for the event. However, the original web page may contain irrelevant images, such as advertisement images. Hence, it is difficult to exactly parse the images based on the html of the web pages. We propose an alternative solution that we use Google Image Search [3] to search the images by using the news article title as the query. Among the returned images, we select the top-ranked ones as the associated images of the news article.

### 4.2 Experimental Settings

To evaluate the performance of various methods, we conduct experiments on our proposed dataset MidEast-TE-mm, as described in Section 4.1. Consistent with previous methods, we employ the Accuracy (Acc) as the evaluation metric.

*4.2.1 Compared Methods.* In addition to the forecasting methods with LLMs, we also implement a list of representative traditional methods. For traditional methods, only textual modalities are involved in the training process, as these methods are fixed. We train the models on the training set, selecting the best-performing model based on the validation set results, and obtain the final results of the testing set. For LLM-based methods, on the other hand, testing is generally done in a zero-shot manner, *i.e.,* , directly test them on the testing set. The specific methods are shown below:

- **ConvTransE [32]:** The method is a static knowledge graph representation learning technique. It employs both a convolutional neural network and a translational operation to identify patterns within triplet data.
- **RGCN [31]:** RGCN is also a static knowledge graph representation learning approach. It leverages a graph convolutional neural network architecture to capture the diverse relations between entities.
- **RE-GCN [20]:** RE-GCN is a state-of-the-art method for temporal knowledge graph (TKG). It utilizes a combination of graph neural networks and recurrent neural networks to capture both the relational patterns and temporal dynamics within the data.
- **LoGo [27]:** The LoGo method is the current state-of-the-art approach for temporal complex event (TCE), which stands for modelling relationships within and between complex events from both local and global perspectives, respectively.
- **GPT-3.5-Turbo:** The GPT-3.5-turbo model is the latest iteration of the GPT (Generative Pre-trained Transformer) language model developed by OpenAI. It builds upon the capabilities of earlier GPT models, leveraging an enhanced transformer architecture to achieve state-of-the-art performance on a wide range of natural language processing tasks.

---

[3]https://images.google.com/

**Table 1: Performance (accuracy) comparison between zero-shot LLM-based methods and the non-LLM methods in both settings of object entity prediction and relation prediction. For LLM-based methods, we include multiple backbones with two representative forecasting method, *i.e.,* ICL and RAG. Results of our methods are highlighted with grey backgrounds, where the key novelty lies in the design of multimodal model.**

| Model Type/Backbone | Forecasting Model | Multimodal Model | Object Entity Prediction | | Relation Prediction | |
|---|---|---|---|---|---|---|
| | | | Text | Graph | Text | Graph |
| Non-LLM | ConvTransE [32] | Uni-modal | N/A | 0.3737 | N/A | 0.7327 |
| | RGCN [31] | Uni-modal | N/A | 0.3777 | N/A | 0.7203 |
| | RE-GCN [20] | Uni-modal | N/A | 0.3879 | N/A | 0.7333 |
| | LoGo [27] | Uni-modal | N/A | 0.3969 | N/A | 0.7406 |
| Gemini-1.0-Pro-Vision[3] | ICL [16] | MLLM[3] | 0.3023 | 0.3319 | 0.5541 | 0.6085 |
| Gemini-1.0-Pro[3] | ICL [16] | Uni-modal | 0.3312 | 0.3657 | 0.5900 | 0.6257 |
| | | **MM-Forecast (ours)** | **0.3527** | **0.3837** | **0.6087** | **0.6324** |
| | RAG [17] | Uni-modal | 0.3340 | 0.3669 | 0.6081 | 0.5866 |
| | | **MM-Forecast (ours)** | **0.3425** | **0.3692** | **0.6121** | **0.5991** |
| GPT-3.5-Turbo[4] | ICL [16] | Uni-modal | 0.3063 | 0.3431 | 0.4847 | 0.5345 |
| | | **MM-Forecast (ours)** | **0.3414** | **0.3522** | **0.5317** | **0.5521** |
| | RAG [17] | Uni-modal | 0.3272 | 0.3397 | 0.4943 | 0.4666 |
| | | **MM-Forecast (ours)** | **0.3652** | **0.3647** | **0.5152** | **0.5113** |

- **Gemini-1.0:** Gemini-1.0 is a cutting-edge family of multimodal models developed by the Gemini Team at Google. It is designed to excel in understanding and generating content across various modalities, including text, images, audio, and video.

*4.2.2 Implementation Details.* During the construction of the dataset, the extraction of sub-events is accomplished by a collaborative effort involving both the Gemini-1.0-Pro and GPT-4 models. Then the Gemini-1.0-Pro-Vision model is used to complete the image function identification and subsequent key event selection and complementary information extraction. To ensure the reproducibility, we fixed the temperature parameter to 0 and set the seed parameter to a constant value. When making forecasting, we limit the maximum token length to 256 to prevent invalid responses. To ensure fairness across the experiments, the length of history that can be retrieved is set to 30. Notably, the retrieval models employed included: BM25 [30], Contriever [12], and LlamaIndex [24]. Additionally, considering the limitation of the context window, we further restricted the maximum number of sub-events in the historical context to 50. The specific prompt used in the experiments can be found in supplementary material.

## 4.3 Performance Comparison (RQ1)

We analyze our model's performance, by comparing various baseline methods with our method among various experiment settings, including different formats of input historical events, forecasting models, and forecasting objectives.

*4.3.1 Performance w.r.t. Various Settings.* The overall performance comparison is presented in the Table 1. To comprehensively explore and evaluate the performance of methods, we conduct experiments across multiple dimensions, including the format of data representation (Text of Graph), the construction of historical information

[3]https://ai.google.dev/models/gemini
[4]https://platform.openai.com/docs/models/gpt-3-5-turbo

**Table 2: The study of using different retrieval models.**

| Retriever | Gemini-1.0-Pro | GPT-3.5-Turbo |
|---|---|---|
| **BM25** [30] | 0.3272 | 0.3318 |
| **Contriver** [12] | 0.3335 | 0.3431 |
| **LlamaIndex** [24] | 0.3425 | 0.3652 |

(RAG-based or ICL-based), and the prediction objective (Object or Relation). Clearly, we have the following observations.

First, enhancing LLM-based methods with visual information significantly improves their accuracy across all experimental settings. This demonstrates that our proposed MM-Forecast makes effective use of visual information, leading to a better contextual understanding of historical information. Hence, our method greatly strengthens the inference ability of LLM and makes more accurate event forecasting.

Second, although the performances of all LLM-based methods have been improved, they still under-perform to traditional Non-LLM based methods. The reason is that LLM-based methods are tested in zero-shot manner, while the Non-LLM methods, which follow supervised learning, are still competitive. Notably, by using our MM-Forecast method, LLM-based methods can achieve similar or even better performance than Non-LLM methods for the object entity prediction task.

Third, the relation prediction task exhibits higher absolute performance compared to the object entity prediction task. This suggests that forecasting entities may be more challenging than forecasting relations. There are a few potential reasons for this. First, the set of entity types is much larger than the set of relation types, so predicting specific entities is inherently more difficult given the larger candidate pool. Second, we deem that the information implied in entities is more explicit. Thus when two entities are given for a



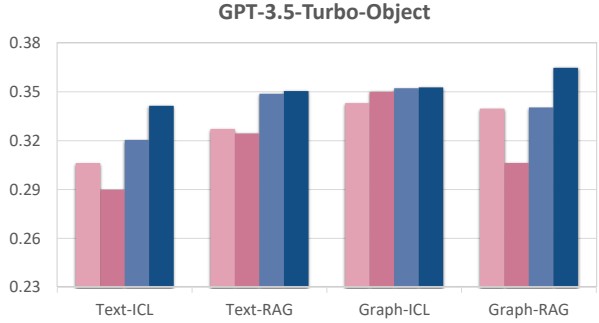

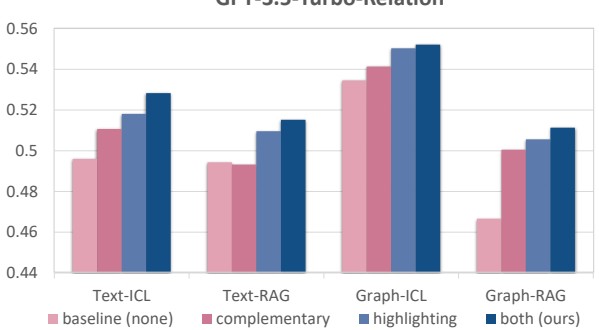

Figure 3: Ablation study of each type of image functions.

relation prediction, it is easier than when the subject and relation are given for an object prediction.

*4.3.2 Performance w.r.t. Directly Using Images.* To illustrate the limitations of existing MLLMs in the task of temporal event forecasting, we also conduct experiments using the Gemini-1.0-Pro-Vsion model [35] directly with images as sub-events. Specifically, this approach leverages the visual processing capabilities of the Gemini-1.0-Pro-Vision model, which embeds image patches as features and seamlessly concatenates them with textual features (for details prompts please refer to the supplementary file). Gemini-1.0-Pro-Vision is a member of the Gemini-1.0 family, and compared to the Gemini-1.0-Pro model, it just has more visual information processing capabilities. From Table 1, we can observe that the accuracy of using images directly is not only much lower than our MM-Forecast, but even worse than that of the method using only textual data. This illustrates the difficulty of existing MLLMs to make effective event forecasting with multiple images, and also reflects the superiority of our MM-Forecast.

*4.3.3 Performance w.r.t. Various Retrieval Models.* The choice of retrieval model can have a significant impact on performance. Since the structured approach employs retrieval based on structured forms, the experiments here involve only unstructured event forecasting. To explore this, we evaluate three different retrieval models, *i.e.,* BM25 [30], Contriver [12], and LLamaIndex [24]. From the results in Table 2, we can observe that the performance progressively improves by using stronger retrieval models, with LLamaIndex performing the best, followed by Contriver, and then BM25. There results verify that stronger retrieval capabilities lead to better forecasting performance, suggesting that retrieval-oriented method

Table 3: The accuracy of image function identification.

| Data-Type | GPT-4-Vision | |
| --- | --- | --- |
| | Text | Graph |
| **Highlighting** | 0.68 | 0.68 |
| **Complementary** | 0.88 | 0.93 |

Table 4: Result comparison between using our identified and randomly-assigned image functions.

| Model | Settings | Object | | Relation | |
| --- | --- | --- | --- | --- | --- |
| | | Text | Graph | Text | Graph |
| GPT-3.5-Turbo | **Random** | 0.3284 | 0.3394 | 0.5156 | 0.5249 |
| | **Ours** | 0.3414 | 0.3522 | 0.5317 | 0.5521 |

design, such as the RAG approach, is a promising direction for future research.

### 4.4 Ablation Study of the Image Functions (RQ2)

To investigate the functions of images at a fine-grained level, we conduct separate ablation experiments for the highlighting and complementary function of images. The results are shown in Figure 3. First, the model that leverages both the highlighting of key events and the complementary information performs the best across the experimental settings. In addition, the performance of the model with only key events highlighted is sub-optimal. This illustrates the effectiveness of the highlighting function of images and the fact that highlighting and complementary reinforce each other to achieve even better prediction results. Second, we can observe that in some settings, the performances of the model with only complementary information are even worse than the baseline model. The possible reason for this is that the providing of complementary information also introduces more noise and therefore leads the degradation of performance. With the performance improving again under the function of highlighting, there is also reflect this reason. Third, comparing the object entity prediction task, the performance of the RAG-based method for the relation prediction task is obviously worse than the ICL-based method. As mentioned in section 4.3.1, the relation prediction is easier compared to the object entity prediction. Therefore, we deem that ICL-based historical event contain enough information to make accurate relation prediction, whereas retrieval may not retrieve relevant information instead.

### 4.5 In-depth Analysis of the Image Function Identification (RQ3)

Improvements in prediction accuracy alone are not enough to fully validate whether images are indeed fulfilling their highlighting and complementary functions. Therefore, we design additional experiments at the data level and prompt level to further confirm the role of images. To verify the correctness of our image functions classification, we randomly sample 100 images of two categories respectively, and then judge the correctness of the classification by the powerful multimodal comprehension ability of the GPT-4-Vision. As shown in Table 3, both classification show high accuracy.

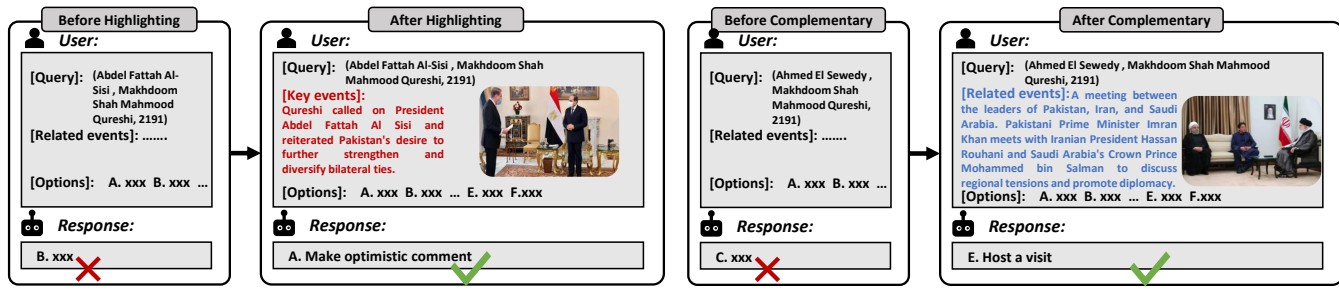

**Figure 4: Case study: two examples that when considering *highlighting* and *complementary* functions of images, our method yields better forecasting results compared with the baselines.**

The lower accuracy of "Highlighting" should be due to its more strict definition. This indicates that the images we used can indeed play the highlighting and complementary functions. In addition to, we conduct experiments where we intentionally include randomly selected sub-events in the prompt, instead of the true key events and complementary information. As shown in Table 4, this random selection of sub-events leads to a decrease in prediction accuracy, indicating that we have indeed identified the true key events and complementary information.

Finally, to provide a visual illustration of the image-text relation, we present two specific examples in Figure 4. The first image emphasizes the event of Makhdoom Shah Mahmood Qureshi's visit to Abdel Fattah Al-Sisi, highlighting their efforts to strengthen and diversify bilateral relations. This highlighting function led to a successful prediction of the event relation. The second image provide supplementary information about the meeting between the two individuals, enabling an accurate prediction of the query. These examples explicitly demonstrate the effect of the image functions on the temporal event forecasting task.

## 4.6 Comparison of Zero-shot and Fine-tuned LLMs (RQ4)

To further explore the potential of our approach on LLMs, we also conduct experiments with open-source LLMs. Specifically, we select one of the most popular open-source LLMs, *i.e.,* Vicuna-7b, to test and further fine-tune it using instruction tuning with QLoRA [9]. The results are presented in Table 5, which also includes the best results for proprietary LLMs and non-LLM methods. We observe that the zero-shot performance of Vicuna-7B is worse than the proprietary LLMs, owing to the inherent capacity gap. However, in the fine-tuned setting, Vicuna-7B achieves substantial performance gains, not only surpassing the proprietary LLMs but also outperforming all the non-LLM methods. These results demonstrate the significant potential of fine-tuning LLMs for the temporal event forecasting task. Leveraging the powerful capabilities of LLMs, with appropriate fine-tuning, represents a promising direction for advancing the state-of-the-art in this domain.

## 5 CONCLUSION AND FUTURE WORK

In this paper, we first proposed the methodological paradigm of multimodal temporal event forecasting and systematically evaluated

**Table 5: Performance of fine-tuned LLMs and its comparison with proprietary LLMs and non-LLM methods.**

|  | Model | Vicuna-7b | LLM | Non-LLM |
|---|---|---|---|---|
| zero-shot | **MM-Forecast-text-h** | 0.2723 | 0.3527 | N/A |
|  | **MM-Forecast-graph-h** | 0.2502 | 0.3837 | N/A |
| fine-tune | **MM-Forecast-text-h** | 0.4490 | N/A | N/A |
|  | **MM-Forecast-graph-h** | 0.5480 | N/A | 0.3969 |

the effects of visual information on the task of temporal event forecasting. Specifically, we first identified two essential functions that images play in the scenario of temporal event forecasting, *i.e.,* highlighting and complementary. Then, we introduced MM-Forecast, a novel framework that leverages visual information to enhance temporal event forecasting. By recognizing the highlighting and complementary functions of images and translating them into verbal descriptions, we were able to seamlessly integrate this visual information into LLM-based forecasting models. Ultimately, this enabled the integration of visual information to enhance temporal event forecasting task. To comprehensively evaluated our proposed approach, we also have designed a series of event forecasting models with different settings, including: different formats of input historical events, forecasting models, forecasting objectives, and backbone LLMs. By implementing these model settings, we obtained a comprehensive understanding of the potential of multimodal event prediction and the importance of leveraging multimodal information for augmentation in temporal event forecasting.

Looking ahead, there are numerous avenues for future work to address the key challenges that have been identified. In particular, we would like to highlight three distinct aspects that warrant further exploration. First, multi-images relationship need to be considered. There are inherent relationships between images in related historical events, and these relationships are also important for event forecasting. Second, seeing is believing. Images have significant effects on the event forecasting task rather than accuracy improvement, that is credibility or trustability. Predictions that are corroborated by images are more likely to be trusted. Third, our current solution is still a multi-step pipeline, while devising an end-to-end approach using MLLMs is intriguing to explore in the future.

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
