# OpenReview forum: "MM-Forecast: A Multimodal Approach to Temporal Event Forecasting with Large Language Models"
_acmmm.org/ACMMM/2024/Conference — MM2024 Poster_

### Official Review · Reviewer_Hig7 · 2024-04-28

**Rating:** 3
**Confidence:** 3

**Summary:**

(1) This is the first comprehensive investigation into the integration of visual information for temporal event forecasting in the era of LLMs.
(2) This paper identifies the function of images within the context of temporal event forecasting and designs an overall framework to recognize and integrate this visual information into LLM-based forecasting models.

**Strengths:**

This paper identifies the function of images within the context of temporal event forecasting and designs an overall framework to recognize and integrate this visual information into LLM-based forecasting models.

**Limitations:**

(1) Compared to the original method, this method converts unimodal into multimodal. While this approach significantly increases computational requirements, it does not result in a substantial performance improvement.
(2) Since the input-output has changed from text to a combination of image and text, it would be worthwhile to consider fine-tuning models such as BLIP-2 and OFA. These models are specifically designed to handle multimodal inputs and outputs, integrating both visual and textual data. Fine-tuning BLIP-2 and OFA could leverage their advanced capabilities in understanding and generating multimodal content, potentially leading to more significant improvements in performance compared to the current approach. By aligning the model's training more closely with the nature of the new input-output format, we may achieve better results in both accuracy and efficiency.
(3) The dataset has not been disclosed, and it is also unknown whether it will eventually be disclosed.
Based on the opinions above, I believe the comparative approach is not comprehensive. Additionally, if the dataset is not published, it will be difficult for others to use this method, making it less referential. Therefore, I give it a Borderline Reject rating.

**Suitability:**

2

---

### Official Review · Reviewer_XdeJ · 2024-05-10

**Rating:** 2
**Confidence:** 3

**Summary:**

In this paper, the authors propose a pipeline to enhance temporal event forecasting by identifying two key functions that images serve in this context: highlighting and complementary. They introduce an Image Function Identification module to determine these functions and this module employs multimodal large language models (MLLMs) to translate these functions into verbal descriptions. These descriptions are then integrated into large language model-based forecasting models to improve their performance. To validate this approach, the authors created a new multimodal dataset, MidEast-TE-mm, which extends an existing event dataset, MidEast-TE, with images.

**Strengths:**

1. The identification of highlighting and complementary functions of images and their integration into forecasting models leverages the contextual power of images to enhance predictive accuracy.

2. The development of the MidEast-TE-mm dataset is an addition to the field, providing researchers with new tools to test and refine multimodal forecasting methods.

**Limitations:**

1. The framework's reliance on images as external knowledge restricts its applicability in real-world settings where such visual data may not be readily available. Additionally, the selection of images may significantly influence the framework's performance, potentially limiting its generalizability and utility across diverse applications. A discussion on how different images affect the outcomes and potential strategies to mitigate these effects would be valuable for understanding the framework’s robustness.

2. The novelty of the proposed framework is questionable, as it does not introduce new techniques and relies on existing methods for converting images to text as external knowledge. This approach does not advance the current state of technology in a meaningful way.

3. Despite utilizing additional resources, including larger pre-trained VLLMs and LLMs, as well as incorporating image information, the framework's performance does not surpass that of traditional methods. This outcome significantly undermines the contribution of the proposed framework.

**Suitability:**

3

---

### Official Review · Reviewer_iep7 · 2024-05-14

**Rating:** 5
**Confidence:** 1

**Summary:**

This paper proposes MM-Forecast, a multi-modal framework for temporal event forecasting from image and text. The authors identify 2 types of functions that images play in news: highlighting and complementary, and use the proposed MM-Forecast framework to incorporate the information from the images into LLM-based forecasting models.
To conduct experiments the authors extend the MidEast-TE dataset to a multimodal version MidEast-TE-mm by adding images relevant to the news obtained from Google Image Search. Extensive experiments are conducted on both structured and unstructured forecasting with both ICL and RAG methods to verify the effectiveness of image information and MM-Forecast method.

**Strengths:**

1.	The topic of temporal event forecasting is important and interesting, and adding multi-modal information sounds reasonable.
2.	Experiments prove that MM-Forecast framework works for many different LLM-based temporal event forecasting methods.

**Limitations:**

1.	In line 535, It may not be a good idea to use images from search engines instead of the news website itself. Images retrieved using news titles may not be closely related to the details of the news text, and the relations between (retrieved images, news text) may be different from the relations between (original images, news text). Do you use any methods to check the quality and relatedness of retrieved images? Why not try using the images from the original web page and use semantic similarity (e.g., CLIP score) to filter out the irrelevant ones?

**Suitability:**

3

---

### Official Review · Reviewer_emNg · 2024-06-09

**Rating:** 3
**Confidence:** 4

**Summary:**

This paper describes the methodology to employing multimodal understanding for temporal event forecasting with the use of LLMs. The aim is to imbue the retrieval and analysis with complementary and highlighting information that includes both textual and visual components to better predict events and perform forecasting. The paper also introduces a new dataset called "MidEast-TE-mm" that extends from the original MidEast-TE dataset to evaluate the proposed approach. The paper utilises a series of forecasting modelling approaches including convolution-based ConvTransE, graph-based RGCN, LLM-based GPT-3.5-Turbo etc. to compare and evaluate the experimental settings.

**Strengths:**

The authors proposed a unique approach to temporal event forecasting methods through a multimodal framework which is currently understudied. Some of the following strengths of the paper include:
- Systematic frameworks to tackle both Unstructured and Structured data sources, and how one can go about constructing a historical event schema that will be important for downstream In-Context Learning.
- The study of applications of LLMs in zero-shot/few-shot event analysis using decomposable structured event representations with clear language
- Image-enhanced retrieval augmented generation to improve prediction performance with real-world data types (news content articles - visual, text, video, audio etc.). A promising approach to future scalability into multimodal methods.

**Limitations:**

With its strengths and potential, there are however some glaring weaknesses that leaves the reader with a lot of question marks on the implementation. The summary of some of the limitations are as follows:
- Severe lack of technical details on implementation. Some of the working blocks including the formulation of the structured event representation, historical event graph, identifying relations between sub-events, the image identification function using Gemini-1.0-Vision-Pro etc. (Supplementary file may be missing in author reviewing package) Some of these critical modules are scantly touched on, leaving a lot of assumptions to be made.
- Functionalities to identify "Highlighting" and "Complementary" not fully descriptive. Still do not understand how is this being done? How does Gemini-Pro transform this information into "verbal descriptions" to be integrated into LLM-based event forecasting.
- Some of the ablation studies did not fully compare across different experimental settings i.e. Comparison of ZS and FT-ed LLMs only considered Non-LLM methods for Object-Entity Prediction (Graph method) but did not include evaluation on Relation Prediction? What about the proprietary LLMs that the paper is comparing to? Which are the LLMs used and evaluated.
- In the FT approach, do the authors consider FT-ing the retrieval modules as well e.g. BM25, Contriever etc?
- In the evaluation of prediction capabilities for forecasting, one of the key issues is in the noise retrieval due to entity/relation mismatches. I would like to understand why the authors not consider the F1-metric which includes the recall on positive samples to understand the true performance of the forecasting framework?

**Suitability:**

3

---

### Meta-Review · Area_Chair_dCS3 · 2024-07-02

**Recommendation:** Accept (Poster)
**Confidence:** 4

**Metareview:**

This paper investigated how images help with temporal event forecasting and how to integrate images into the LLM-based forecasting framework. This paper received 1 weak reject, 2 borderline reject, and 1 weak accept as initial ratings. After the rebuttal period, the 2 borderline reject were increased to 1 weak accept and 1 borderline accept. The remaining main concern from Reviewer XdeJ is the reliance on images as external information. After checking the authors' rebuttal, AC thinks this main concern is addressed and agrees with the authors that the inclusion of images expands its applicability for news articles. Therefore, AC recommend acceptance.

---

### Meta-Review · Senior_Area_Chairs · 2024-07-10

**Recommendation:** Accept (Poster)
**Confidence:** 4

**Metareview:**

This paper received mixed ratings initially. After rebuttal, most reviewers are satisflied with the response and tend to accept the paper, while one reviewer still questioned that some issues are not well addressed. SAC and AC carefully check the paper, reviewes and rebuttal and recommend acceptance of the paper.